# Electrotransfer of IL-15/IL-15Rα Complex for the Treatment of Established Melanoma

**DOI:** 10.3390/cancers12103072

**Published:** 2020-10-21

**Authors:** Shawna A. Shirley, Cathryn G. Lundberg, Richard Heller

**Affiliations:** 1Frank Reidy Research Center for Bioelectrics, Old Dominion University, Norfolk, VA 23508, USA; sashirle2@gmail.com (S.A.S.); clundber@odu.edu (C.G.L.); 2Department of Medical Engineering, University of South Florida, Tampa, FL 33512, USA

**Keywords:** IL-15, IL15Rα, gene therapy, immunotherapy, melanoma, cytokines

## Abstract

**Simple Summary:**

The stimulation of the immune system through the administration of immunomodulatory agents such as cytokines has the potential to be an effective anti-cancer therapy. Obtaining the correct dose is an important aspect with respect to minimizing toxicity and obtaining the desired effect. A method to decrease the toxicity of this type of treatment is to replace the high-dose recombinant protein injections by using DNA expressing genes for one or more of these anti-cancer agents. In this current study, we have evaluated the delivery of interleukin-15 and its receptor in the form of plasmid DNA in a mouse melanoma model. We utilize a delivery approach that can deliver plasmid DNA in a manner that results in the desired level of expression being produced and induces a potent anti-tumor response as well as an immune memory response.

**Abstract:**

Gene electrotransfer (GET) is a safe, reliable, and effective method of delivering plasmid DNA (pDNA) to solid tumors. GET has been previously used to deliver interleukin-15 (IL-15) to mouse melanoma, resulting in long-term tumor regression and the survival of a percentage of treated animals after challenge. To enhance this effect, we evaluated modulating the expression levels of IL-15 and co-expressing its receptor, IL-15Rα. GET was used to deliver plasmids encoding IL-15 and IL-15Rα to established B16.F10 tumors on days 0, 4, and 7. Two delivery protocols that yielded different expression profiles were utilized. Mice that were tumor-free for 50 days were then challenged with B16.F10 cells on the opposite flank and monitored for an additional 50 days. The amount of IL-15 expressed and the presence or absence of IL-15Rα in the treated tumors did not significantly affect the tumor regression and long-term survival. Upon challenge, however, low levels of IL-15 were more protective and resulted in a greater production of anti-tumor cytokines such as IFN-γ and MIP-1β and a greater amount of CD11b+ and CD3e+ cells infiltrating tumors. While mice with high levels of IL-15 showed CD11b+ and CD3e+ cell infiltrate, there was a substantial presence of NK cells that was absent in other treated groups. We can conclude that the level of IL-15 expressed in tumors after GET is an important determinant of the therapeutic outcome, a finding that will help us finetune this type of therapy.

## 1. Introduction

Interleukin-15 (IL-15) is an important mediator of immune function. It is a pleiotropic cytokine related to IL-2 that regulates the function of a variety of cells by signaling through the β and γ chain of the IL-2 receptor [1]. IL-15 is stabilized by binding to its high-affinity alpha receptor (Rα), which is structurally related to IL-2Rα [2]. The ability of IL-15 to regulate and stimulate both innate and adaptive immune cells makes it an attractive anti-cancer agent [3,4,5,6]. Specifically, its ability to recruit, maintain, and stimulate CD8+ T cells, NK cells, and NKT cells within the tumor itself would be a significant advantage to overcoming immune evasion.

IL-15 is highly regulated and its expression is tightly controlled [7]. Binding to its high-affinity alpha receptor stabilizes the protein, prevents degradation, and increases bioactivity [8,9,10,11,12,13]. The IL-15/IL-15Rα complex formation is important for trans-presentation to target cells and subsequent signaling though the βγ receptor [14,15]. The delivery of genes encoding a soluble receptor α along with IL-15 could improve the efficacy of this anti-cancer immune therapy. Previously, we demonstrated that the electrotransfer of IL-15 directly to tumors leads to the regression and long-term survival of mice with melanoma [16,17]. Rowley et al. [18] explored the use of IL-15 together with Il-15Rα by transducing tumor cells with a retrovirus encoding IL-15 linked to IL-15Rα. Following the injection of the transduced tumor cells, a slowing of tumor growth as well as an enhanced immune response was observed. An additional study transfecting unstimulated CD8+ T-cells with RNA encoding IL-15Rα using nucleofection (electroporation) followed by exposure to IL-15 resulted in increased viability and proliferation. This was also observed when transfecting unstimulated T-cells with IL-15 linked to IL-15Rα which enhanced this effect further, as well as after cells were injected into mice [19]. While this previous study successfully demonstrated the potential of combining IL-15/IL-15Rα, here we seek to improve on this therapy by delivering both IL-15 and IL-15Rα using electrotransfer protocols and to test the concept further by treating established tumors and evaluating whether the regression of existing tumors could be achieved.

Gene electrotransfer (GET) is an efficient, safe, and effective method used to physically deliver DNA to tissues [20,21]. GET is mediated by the generation of electric fields in tissue through the administration of electric pulses via electrodes placed around the tumor. The level of gene expression generated can be controlled based on the selection of electrode and pulse parameters [22,23,24,25,26,27]. We previously demonstrated that achieving the appropriate level of cytokine expression locally is important in generating the most effective immune response, resulting in tumor regression and long-term survival [23].

In the current study, in order to augment the anti-tumor effects previously obtained several aspects of this immunotherapy approach were examined. A major emphasis of the current study was to evaluate the delivery of IL-15Rα together with IL-15 either on separate plasmids or together on one plasmid. Three plasmids were utilized to test this approach. One plasmid encoded only IL-15 (pAG170), a second plasmid encoded only IL-15Rα (pAG115), and one plasmid encoded both IL-15 and IL-15Rα using a dual promoter (pAG208). One aspect that was explored in this current study was the impact of expression levels on efficacy. Therefore, two GET delivery protocols were utilized in order to achieve different levels of IL-15 following the delivery of the different plasmids. The evaluation of these protocols included level of response against the treated tumor and the ability to induce protection following challenge with a second injection of tumor cells.

## 2. Results

Previous studies from our lab showed that IL-15 GET delivered to established tumors in a mouse melanoma model resulted in complete tumor regression [16,17]. Systemic immune effects of the treatment, evidenced by protection from challenge, were present in 45% of the treated animals. In this study, we evaluated the delivery of IL-15 alone or in combination with its high-affinity receptor IL-15Rα to potentially improve the therapeutic outcome. The two GET protocols used were EP1 and EP2. EP1 (six rotating pulses of 1300 V/cm, 100 µs duration, delivered using a circular six-needle array) has been shown to generate relatively low levels of gene expression, while EP2 (10 unidirectional pulses of 600 V/cm, 5 ms in duration, delivered using parallel plate applicator) has been shown to generate high levels of gene expression. Both protocols are widely used for GET in preclinical and clinical studies. Experiments were designed to determine the optimal electrotransfer parameters that would deliver the most appropriate IL-15 levels to improve the therapeutic outcome. Work was performed in a mouse melanoma model. B16.F10 cells were injected into the flank of C57Bl/6 mice. Once the tumors reached approximately 4–5 mm in diameter (6–7 days), treatments were initiated.

### 2.1. IL-15 and IL-15/IL-15Rα Complex Expression Following Intratumor Delivery

The initial experimental approach was designed to determine the expression levels that could be obtained following direct delivery to tumors. Three plasmids and two GET protocols were evaluated. The three plasmids utilized were a dual-promoter plasmid encoding IL-15 and IL-15Rα (pAG208), along with plasmids encoding IL-15 (pAG170) and IL-15Rα (pAG115). The two GET protocols utilized were EP1 and EP2, along with two electrode configurations. The needle electrode applicator (N) consisted of six needles in a circular array with an outer diameter of 1 cm. Only four of the six needles were active for each pulse, and the field was rotated 60° after each pulse so that at the end of the six pulses a full 360° circle was completed [27]. The gap between the active electrodes was 0.92 cm. The caliper electrode (C) consisted of two flat stainless-steel plates (5 mm × 5 mm) mounted on the end of a Vernier caliper. The two plates were placed on either side of the tumor prior to administering pulses, the gap was measured, and the applied voltage was set according to the gap.

The plasmids were injected directly into the tumor at a volume of 50 µL and a concentration of 1 mg/mL. The three plasmids were evaluated individually and pAG115 and pAG170 were also delivered together using equal volumes of both. The first evaluation was to test the expression levels of IL-15 following delivery at several time points from 6 h to 8 days. Delivery was performed on days 0, 4, and 7. The highest expression was observed when pAG170 was delivered with EP2. Elevated expression was also seen when pAG170 was delivered with EP1. In both cases, the expression dropped significantly after day 1 (Figure 1A). Subsequent deliveries did not restore the expression levels, which may have been due to the regression of the tumors. With respect to pAG208, the dual plasmid expression was elevated with EP2, though not to the extent as that obtained with pAG170, and although there was a relative decrease from the initial levels, the expression was maintained longer with this plasmid. The delivery of pAG208 with EP1 had only a slight elevation of the expression levels. The delivery of pAG170 together with pAG115 resulted in expression levels similar to those of pAG170 alone for each of the two GET conditions; however, the expression was for a longer period, even though there was also tumor regression. The results observed with pAG208 and pAG170 together with pAG115 are interesting, as they support the notion that the half-life of IL-15 can be extended when complexed with IL-15Rα.

The levels of the IL-15/IL-15Rα complex following delivery were also explored. The same experimental setup as described above was performed. Not surprisingly, the presence of the complex was only seen after the delivery of the dual plasmid or when both pAG170 and pAG115 were delivered together. The highest and most sustained levels were seen when pAG170 and pAG115 were delivered together using EP2 (Figure 1B). High levels of the complex were observed with EP1, but they dropped off after day 1. There was a low to moderate presence of the complex after the delivery of pAG208 with EP2, which was maintained for 5 days, and the levels were even lower when the delivery was performed with EP1.

### 2.2. Tumor Regression and Long-Term Survival after IL-15 GET

Plasmids were delivered to established tumors using either EP1 or EP2 as a series of three treatments on days 0, 4, and 7. The tumor volume was measured over the course of nine weeks, as shown in Figure 2. The injection of plasmid with no electrotransfer pulses was used as a control. All the animals that received this treatment showed no tumor regression. The delivery of pAG208 using EP1 resulted in complete regression in only 60% of the treated animals. This is less than when EP1 was used to deliver both pAG115 and pAG170 and pAG170 alone, which showed complete regression in 100% and 90% of the treated animals, respectively (Figure 2). Once the tumors regressed, the animals remained tumor-free for the duration of the 9-week observation period. The delivery of pAG115 with EP1 showed no regression in any of the treated animals. The delivery of pAG208 using EP2 resulted in complete regression in 90% of the animals compared to 60% and 70% of animals treated with both pAG115 and pAG170 or pAG170 alone, respectively (Figure 2). From these results, it appears that the selection of electrotransfer protocol plays a role in the tumor regression and overall survival (Figure 3). For example, the delivery of pAG208 with EP2 resulted in higher levels of expression and a higher percentage of complete tumor regressions than delivery with EP1. In contrast, the delivery of pAG170 or the combination of pAG170 and pAG115 with EP2 again resulted in a higher expression; however, a better tumor response was obtained with EP1. Overall, pAG208 resulted in a lower expression than the delivery of the other plasmids, but high levels of tumor regression were observed when delivered with EP2. It is possible that the high levels of expression achieved with EP2 when delivering pAG170 or pAG115 together with pAG170 may not have induced a sufficient anti-tumor immune response. With respect to IL-15Rα, these data do not suggest a clear advantage for selecting to deliver IL-15 with IL-15Rα rather than IL-15 alone.

### 2.3. Delivery of IL-15/IL-15Rα Protects a Greater Percentage of Mice from Challenge

The animals that were tumor-free after the treatment described in Section 2.2 were challenged subcutaneously with B16.F10 cells on the opposite flank and closely monitored for tumor development. Animals that developed tumors post-challenge were termed non-resistant and those that did not develop tumors were termed resistant. Table 1 shows the number of animals that were resistant to challenge after 50 days, as well as the overall percentage of the original number that were tumor-free after treatment for the duration of the entire experiment (primary and challenge). Resistance to challenge indicates the generation of a systemic immune memory response.

The delivery of IL-15/IL15Rα (pAG208) protected more animals from developing a second tumor than the delivery of IL-15 alone (pAG170) or the delivery of both genes on separate plasmids (pAG115 and pAG170). Using EP1 to deliver the plasmids showed only a slight advantage in using IL-15/IL-15Rα over IL-15 alone by protecting 40% of the animals from challenge compared to 30%, respectively. When both genes were delivered on separate plasmids, only 20% of the animals were protected. Using EP2 to deliver the plasmids had a more definitive outcome. In this case, delivering pAG208 was able to protect 70% of the animals from developing a second tumor, the highest level of protection for all groups tested. The delivery of IL-15 alone or both genes on separate plasmids using EP2 was not able to confer protection to any of the treated animals. From these results, it appears that different combinations of plasmid and electrotransfer parameters may be inducing different cellular responses within the tumor. It is also important to note that the delivery of pAG208 with EP2 did not result in high levels of IL-15 or IL-15/IL-15Rα complex. However, this did result in the highest level of protection from challenge.

### 2.4. Cytokines Generated in the Tumor after GET

In order to explore how immune activation may be involved in the observed tumor regression and long-term systemic response, a panel of cytokines was measured using a magnetic bead multiplex assay. The cytokines included in the multiplex were: IFNγ, IL-1α, IL-6, IL-10, IL-12 p40, IL-12 p70, IL-15, IP-10, MCP-1, MIP-1β, and TNFα. Tumors were collected at various time points over the course of treatment and assayed for cytokine levels. Other than IL-15, only IFNγ, IL-1α, IL-6, and MIP-1β showed variation between the treatment groups. The results for these four cytokines are shown in Figure 4. The groups chosen for analysis were based on the percentage of overall survival after treatment and challenge. Since the delivery of both genes on separate plasmids did not induce a robust systemic response, they were not included in this analysis. Direct comparisons are made for groups that received both genes on a single plasmid and groups that received IL-15 only using both electrotransfer protocols. Animals treated with IL-15Rα only using EP2 and pAG208 injection only were included as controls for the purposes of this experiment.

Overall, the choice of EP1 or EP2 to deliver each plasmid did not result in a significant difference in the level of cytokines induced. The expression of IL-15 from pAG208 and pAG170 is driven by different promoters and, as such, the expression levels detected in the tumor are different. IL-15 appears to be generated at higher quantities from pAG170 than pAG208 (Figure 1A). At the early time point, after the first treatment with pAG170 slightly higher levels of IL-1α and IL-6 compared to pAG208 were observed (Figure 4A,B). After the second treatment, the groups treated with pAG208 showed higher levels of IFN-γ and MIP-1β compared to pAG170 (Figure 4C,D). This indicates there may be a more cellular, non-specific response to the delivery of pAG170, while the delivery of pAG208 stimulates a more specific response.

### 2.5. GET Immunotherapy Generates Local Lymphocytic Infiltrate

In order to determine what effects the therapy had on the local tumor environment, tumors were collected shortly after the first and last treatments (day 1 and day 9) and examined by immunohistochemistry. Only one electrotransfer condition, EP2, was selected for this analysis, as it showed protection from challenge in the largest percentage of animals as well as the greatest disparity in protection based on the selection of plasmid.

The presence of an increased lymphocytic infiltrate in the treated tumors could be an indication that tumor regression is immune cell-mediated and not due to strong electric fields generated by the electrotransfer pulses. The histology of tumors collected during the course of treatment showed increased tumor damage when IL-15 was locally expressed compared to the controls (Figure 5). CD11b, CD3e, and CD335 antibodies were used to determine the general populations of monocytes and antigen-presenting cells, T lymphocytes, and natural killer (NK) cells, respectively, present in the tumors after treatment (Figure 6). Compared to non-treated tumors, the plasmid electrotransfer increased the presence of immune cells in the tumor. pAG208 induced more CD11b+ and CD3e+ cells than AG170, but the opposite is true for CD335+ cells. This indicates that, while pAG170 induces cellular infiltrate, the immune response driving tumor regression may be non-specific. The increased presence of CD335 + NK cells and the amount of early tumor damage compared to the other groups are indicators of this. The delivery of pAG208 induces early CD11b+ cell infiltrate that facilitates antigen presentation and late CD3+ cell infiltrate, which suggests the generation of a specific immune response to tumor antigen. The lack of increase in NK cell infiltrate seen with pAG170 also lends itself to the theory of the generation of a tumor-specific response.

### 2.6. Get Immunotherapy Generates a Systemic Response

The GET delivery of pAG208 with EP2 generated the highest level of protection following challenge with an injection of fresh B16.F10 cells. This protocol also resulted in the highest level of local immune response with regards to the infiltrate of T-cells and a cytokine profile indicating a more specific response. Being able to sustain a long-term disease-free state and the generation of local immune response is important in controlling tumor spread; it is also important to determine if a robust systemic response could be generated that could prevent and/or eliminate distant tumors in visceral organs. To test this, we utilized a lung colonization model.

In addition to pAG208 delivered with GET EP2, the groups included pAG170 GET EP1 (best response with that plasmid) and pAG115 delivered with GET EP2 as a control. Tumors were established on the left flank of mice and treated on days 0, 4, and 7. On day 0, the mice received an intravenous injection of B16.F10 cells that had been stably transfected with luciferase. The mice were imaged with an IVIS in vivo imager to detect growth within the lungs. Mice that were treated with pIL-15/IL-15Rα complex had a greater level of protection from secondary tumor formation using a lung colonization model (Figure 7). However, to a lesser extent, treatment using GET EP1 to deliver pAG170 slowed the tumor growth within the lungs.

## 3. Discussion

The administration of cytokines to stimulate an immune-mediated anti-tumor response has been explored for several decades [28,29,30]. Interferon-alpha (IFN-α) was the first cytokine to be approved by the U.S. Food and Drug Administration (FDA), initially for the treatment of hairy cell leukemia in 1986 and then for metastatic melanoma in 1995 [28,29]. Interleukin-2 (IL-2) was also evaluated as a potent stimulator of T-cell activity. High-dose IL-2 received FDA approval for metastatic renal cell cancer in 1992 and advanced melanoma in 1998 [28,29]. Since then, several other cytokines have been tested in a variety of indications, including interferon-gamma, granulocyte macrophage colony-stimulating factor, IL-12, IL-15, and IL-21 [28,29,30]. These cytokines were initially evaluated as single agents. In these trials, while there was some low level of responses, there was also significant toxicity. One drawback of these approaches is the toxicity associated with the systemic administration of these agents. This may have also contributed to the low response rates related to low levels of the cytokines within the tumor environment.

A typical issue with these agents is their short half-life. One approach to potentially overcome this is to administer cytokine therapies at high doses which can result in adverse reactions. To potentially get around this issue, as well as to prolong the presence of the therapeutic agent, a gene-based approach can be used. One example of this approach is the delivery of IL-12 in the form of plasmid DNA (pIL-12). To better control the expression levels and kinetics, it was delivered using gene electrotransfer [23]. Preclinical studies demonstrated the efficacy and safety of this approach [31,32]. In clinical trials testing the approach in melanoma patients, this approach was shown to be effective with no systemic treatment-related adverse effects [33]. In addition, in both Phase I and Phase II trials responses were seen in both treated and untreated lesions, with approximately 15% of the patients achieving durable complete response of all lesions [34]. This approach is being tested with other tumor types, including Merkel cell carcinoma [35].

IL-15 has shared functions with IL-2 and was thought to be an attractive alternative and, unlike IL-2, does not attract T-regulatory cells nor induces activation-induced cell death. However, IL-15 as a monotherapy did not result in a sustained anti-tumor response [28,29]. It was noted that, for IL-15 to be effective, it was necessary for it to be associated with IL-15 Rα [28,36]. Several studies have been conducted utilizing IL-15/IL-15Rα (superagonist). In clinical trials with the IL-15 superagonist, there were higher levels of T-cytotoxic cells in the circulation for a longer period. There was also a higher level of NK cells and T-effector cells. Although it is well established that IL-15 or IL-15/IL-15Rα complex can be a potent stimulator of T cytotoxic and NK cells, high levels of IL-15 or the complex can actually result in reduced proliferation, exhaustion, and reduced anti-tumor activity [37]. A correlation between high IL-15 serum levels, poor prognosis in melanoma patients, and reduced efficacy in treatment with ipilimumab was observed [38]. High-dose IL-15 has also been associated with severe adverse effects. including systemic inflammation [37]. These effects are related to an extended half-life of the IL-15 complex. There are also additional studies being conducted evaluating the potential of engineering cells to express the IL-15/IL-15Rα complex or to deliver via plasmid DNA or viral vectors [39]. These approaches are designed to extend the time that IL-15 is present and to potentially augment the anti-tumor response.

In the study reported here, we set out to answer two questions: (1) Does the delivery of IL-15 along with its soluble receptor IL-15Rα improve therapeutic outcome? (2) Can the selection of electrotransfer parameters enhance this outcome by modulating the level of IL-15 expression? Using GET to deliver IL-15 and the soluble IL-15Rα on a single plasmid (pAG208), we were able to demonstrate an effective local anti-tumor response similar to that obtained using IL-15 only (pAG170). The systemic response generated by pAG208 exceeded that of delivering only pAG170. By selecting appropriate delivery parameters, we were able to enhance the long-term protective effects of this therapy and show protection from challenge in 70% of treated animals, which is an improvement on what we have described previously [16,17]. Interestingly, this was similar to results obtained in preclinical studies in the same model with pIL-12 [22], although the GET parameters to obtain this response were different.

The electrotransfer protocols were selected because they generate different levels of gene expression [23,39]. We sought to determine if increased local IL-15 expression, with or without the presence of IL-15Rα would result in greater anti-tumor response and protection. Due to gene expression being driven by different promoters, pAG208 produced less measurable IL-15 than pAG170 even when EP2 was used for delivery. To account for this discrepancy and examine the effects of the presence of the receptor, IL-15 and IL-15Rα on separate plasmids (driven by the same promoter) were delivered simultaneously. Though there was not an apparent advantage in using IL-15/IL-15Rα over IL-15 alone, or high levels of IL-15 over low levels of IL-15 to induce tumor regression of the treated tumor and long-term survival, it became clear upon challenge that high levels of IL-15 do not protect from challenge and a lower expression induces specific anti-tumor responses. The delivery of both genes on separate plasmids showed similar levels of protection to the delivery of IL-15 alone.

The expression of lower levels of IL-15 in the tumors generated higher levels of the anti-tumor cytokines IFN-γ and MIP-1β than what was generated when higher levels of IL-15 were obtained. IFN-γ and MIP-1β peaked five days into the treatment regimen, creating favorable conditions for the generation of a specific rather than a non-specific immune response that would result in immune memory. The histological examination of treated tumors revealed that high levels of IL-15 induced a robust innate response that appears to be mainly driven by NK cells that resulted in a great amount of tissue damage. The lower levels of IL-15 induced far less NK cell infiltrate but more CD3+ cells. This high level of IL-15 in the tumor resulted in tissue destruction that occurred rapidly, and though antigen-presenting cells may have been present, the low number of T cells in the tumor did not allow for the generation of an adequate memory response.

The results presented in this study as well as by others have shown a clear advantage of utilizing IL-15/IL-15Rα over the monotherapy. In clinical trials evaluating IL-15, a low level of response was observed. To potentially overcome these poor responses, studies are being conducted to determine the potential of combining IL-15 or IL-15/IL-15Rα with other immune modifiers, including checkpoint inhibitors [40,41,42]. Interestingly, in the pIL-12 clinical trials, patients who had a poor response were observed to have elevated tumor levels of PD-1 and PD-L1 [34,43]. A phase II clinical trial combining pIL-12 and anti-PD-1 in immunologically quiescent melanomas resulted in a higher than expected response rate [44], with a complete response rate (all tumors responding) of 36% [44]. Determining the appropriate delivery conditions for a plasmid encoding IL-15/IL-15Rα and combining with a checkpoint inhibitor could result in similar or higher response levels.

## 4. Materials and Methods

### 4.1. Plasmids, Cells, Mice

The plasmids used in this study were gifted to us by Drs. B. Felber and G. Pavlakis (NCI, Bethesda MD, USA) and commercially prepared by Aldevron, (Fargo, ND, USA), with endotoxin levels <100 EU/mg and diluted in sterile saline solution. pAG208 is a dual promoter plasmid expressing mouse IL-15 with a GM-CSF secretory signal sequence from a simian CMV promoter and IL-15Rα from a human CMV promoter. The plasmids pAG170 and pAG115 express IL-15 with a GM-CSF secretory signal sequence and IL-15Rα from a human CMV promoter, respectively. B16.F10 melanoma cells were maintained in McCoy’s 5A media (Mediatech, Manassas, VA, USA) supplemented with 10% fetal bovine serum (Life Technologies, Grand Island, NY, USA) and 1% gentamycin at 37 °C and 5% CO_2_ humidified air. Female C57BL/6J mice (6–8 weeks old) were purchased from Jackson Laboratories (Bar Harbour, ME, USA) and housed in compliance with the guide to the care and use of laboratory animals. All the mouse studies were performed with the approval of the Old Dominion University Institutional Animal Care and Use Committee. Two approvals were obtained. Protocol 11-007 was obtained on 13 February 2015 and Protocol 17-019 was obtained on 23 October 2018.

### 4.2. Tumor Generation

Primary tumors were generated on the shaved left flank of the animal by subcutaneous injection of 50 µl of B16.F10 melanoma cells (1 × 10^6^ cells). After the tumors reached a diameter of 3 to 5 mm, plasmid DNA was delivered using various electrotransfer protocols. Secondary tumors were generated for the purposes of challenge by subcutaneous injection of 5 × 10^5^ B16.F10 cells on the right shaved flank of the animal approximately 50 days after gene electrotransfer. The tumor volume was measured using digital calipers and recorded for each animal twice weekly using the formula for the volume of an ellipsoid: v = πab^2^/6, where a is the long diameter and *b* is the short diameter [45]. Mice whose tumor volumes exceeded 1000 mm^3^ were removed from the study and humanely euthanized.

### 4.3. Plasmid DNA Electrotransfer

A total of 50 µg of plasmid DNA (1 mg/mL) was injected directly into the tumor using a syringe with a 25-gauge needle. The electrode was placed around the tumor and pulses were delivered using an ECM 830 Square Wave Pulse Generator (BTX Harvard Apparatus, Holliston, MA, USA) according to the chosen protocol parameters. The GET protocol EP1 consisted of six rotating 1300 V/cm, 100 µs pulses delivered using an applicator with a circular configuration of 6 penetrating electrodes. The protocol EP2 consisted of 10 unidirectional 600 V/cm, 5 ms pulses delivered using adjustable parallel plate caliper electrodes. The mice received a series of three plasmid deliveries on days 0, 4, and 7.

### 4.4. Cytokine Measurements

Tumors were collected from animals in separate experiments at 6 and 24 h after a single delivery, 24 h after two deliveries, and 24 h after all three deliveries, and then homogenized in cold phosphate buffered saline (Mediatech, Manassas, VA, USA) containing protease inhibitor (Roche, Indianapolis, IN, USA). The homogenate was centrifuged and the supernatant collected and assayed for cytokine expression using a Milliplex MAP kit mouse cytokine magnetic bead panel multiplex (Millipore, Billercia, MA, USA) according to the manufacturer’s instructions.

### 4.5. Immunohistochemistry

Tumors were collected 24 h after a single delivery, two deliveries, and three deliveries of pDNA in separate experiments; embedded in OCT; frozen on dry ice; and stored at −80 °C. Standard protocols were used. Briefly, 7 µm cryosections were prepared and fixed using a 75% acetone and 25% ethanol mix. Sections were incubated in 5–10% animal serum blocking buffer for at least one hour at room temperature. An avidin/biotin blocking kit (Vector Labs, Burlingame, CA, USA) was used to block the endogenous biotin present in the samples prior to the addition of the primary antibody. Antibodies to the markers CD11b (14-0112, 1:200; eBioscience, San Diego, CA, USA), CD3e (14-0031, 1:100; eBioscience), and CD335 (NKp46) (137615, 1:50; BioLegend, San Diego, CA, USA) were used. Biotinylated secondary antibodies anti-rat IgG (13-4813, 1:500; eBioscience) and anti-armenian hamster IgG (13-4113, 1:300; eBioscience) were used, except for the samples stained for CD335, as the primary antibody is biotinylated. Signal was detected using FITC conjugated streptavidin (11-4317, 1:500; eBioscience). DAPI (Sigma, St. Louis, MO, USA) was used as a counterstain to visualize nuclei. Cover slips were mounted using VectaSheild Reagent (Vector Labs) and the images recorded using a DP70 camera attached to a BX-51 fluorescent microscope (Olympus, Center Valley, PA, USA).

### 4.6. Lung Colonization Model

B16.F10 (B16.F10-Luc) cells that are stably transfected to express luciferase were used. This enabled the cells to be tracked noninvasively. Mice with B16.F10 tumors already established on the left flank as described above were injected intravenously into the tail vein with 1 × 10^5^ B16.F10-Luc cells in a volume of 50 µL of sterile PBS on the first day of treatment. Only the subcutaneous tumor was treated and the treatment was as described above. Mice were imaged on days 6, 8, and 11 using an IVIS Spectrum whole body imaging system (Perkin Elmer, Waltham, MA, USA). To obtain an image, luciferin is administered via an intraperitoneal injection at a concentration of 15 mg/mL and a dose of 150 mg/kg weight. For a 20 g mouse, 200 µL is injected. All the animals are imaged within 15 min after the injection of luciferin. The IVIS Spectrum takes an image of the animal and can measure the amount of luminescence being produced at specific sites.

### 4.7. Statistical Analysis

Data were log transformed to achieve normal distribution. Multiple groups were compared using a one-way ANOVA with the Tukey–Kramer Multiple Comparison post-test. Pair-wise comparisons were conducted using a paired *t*-test. The critical alpha for all the comparisons was 0.05.

## 5. Conclusions

From the results obtained in this study, it appears that the level of IL-15 expressed in the tumor has a greater impact on the overall survival and protection from challenge than the presence or absence of the soluble receptor. At higher expression levels, an innate immune response was induced, and while it was sufficient to induce the complete regression of the treated tumor, a robust memory response was not induced, resulting in a reduced protection following challenge and less protection from lung colonization. In addition, though the presence of IL-15Rα in tumors is thought to stabilize the protein and allow for more efficient trans-presentation and signaling on the target cells, from these results we cannot at this time fully discern its role in the responses seen. Further studies need to be done at low levels of expression in the presence and absence of the receptor to tease out its role, if any, in this GET therapy. Additional studies should also be conducted to determine how the presence of checkpoints may influence the response and to test the combination of IL-15/IL-15Rα with checkpoint inhibitors.

## Figures and Tables

**Figure 1 cancers-12-03072-f001:**
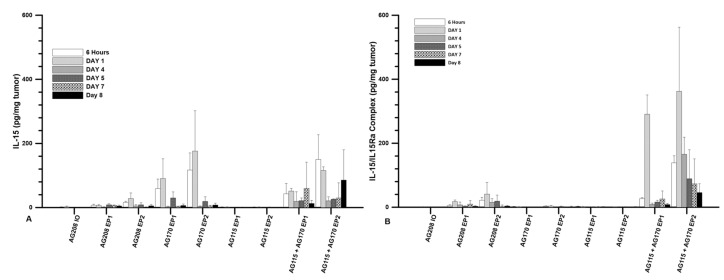
IL-15 and IL-15/IL-15Rα complex expression after GET. B16.F10 tumors were established on the left flank of C57Bl/6 mice. Once tumors reached 3–5 mm in diameter, the intratumor injection of the plasmid was performed as described in methods. Following injection, electrotransfer was performed using either EP1 or EP2. EP1 = 1300 V/cm, 100 us, 6 pulses of EP1 delivered using a 6 penetrating needle array. EP2 = 600 V/cm, 5 ms, 10 pulses of EP2 delivered using caliper electrodes. Level of transgene expression within the B16.F10 tumors was measured by ELISA using tumor homogenate collected at the indicated time points. *n* = 4 for each group. (**A**) levels of Il-15 after delivery; (**B**) levels of IL-15/IL-15Rα after delivery. IO = injection only.

**Figure 2 cancers-12-03072-f002:**
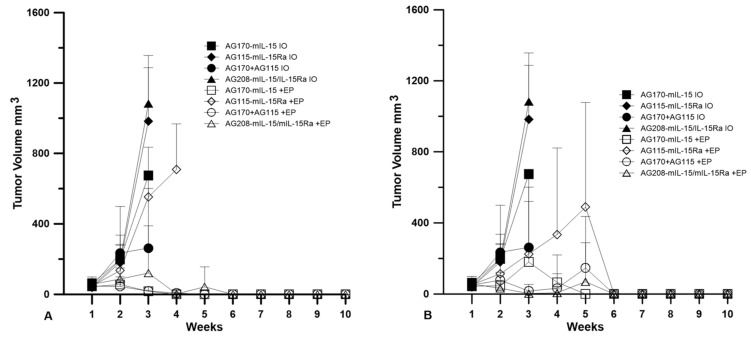
IL-15 GET induces tumor regression. Animals were given a series of three deliveries of intratumoral plasmid DNA using GET on days 0, 4, and 7. (**A**) EP1 delivered using a 6 penetrating needle array and (**B**) EP2 delivered using caliper electrodes. Tumor volume was measured in mm^3^ and monitored weekly over a nine-week period. Data are presented as the mean tumor volume and error bars represent the standard deviation. *n* = 10 for each group. IO = injection only; EP = electrotransfer.

**Figure 3 cancers-12-03072-f003:**
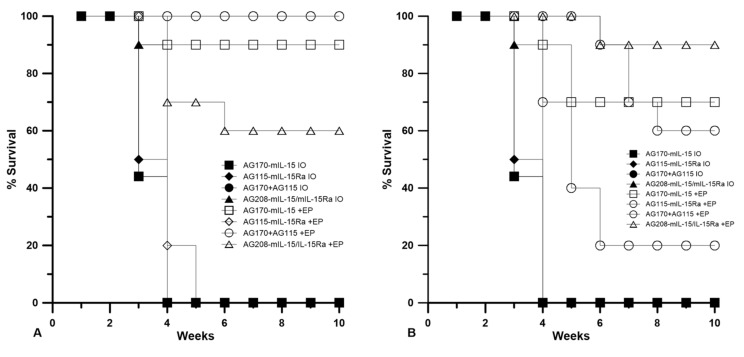
IL-15 GET induces the long-term survival of mice. The survival of mice monitored above is represented as the percent survival of the experimental group over time. Electrotransfer protocols were used as indicated. (**A**) EP1 = 1300 V/cm, 100 us, 6 pulses of EP1 delivered using a 6 penetrating needle array. (**B**) EP2 = 600 V/cm, 5 ms, 10 pulses of EP2 delivered using caliper electrodes. *n* = 10 for each group. IO = injection only; EP = electrotransfer.

**Figure 4 cancers-12-03072-f004:**
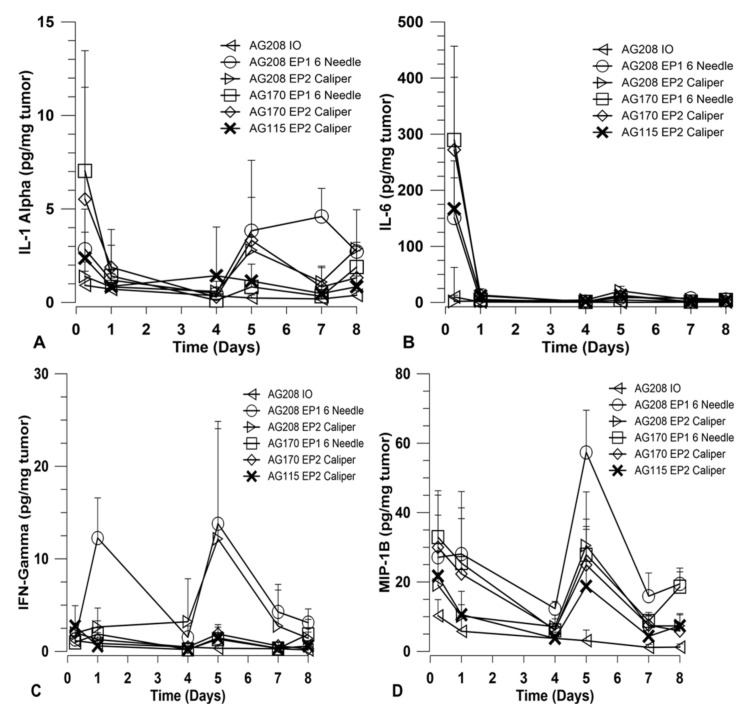
IL-15 GET induces cytokine in the tumor microenvironment. Cytokine levels were measured using a multiplex assay from tumor homogenate collected at various time points during the treatment protocol. Cytokines included in the multiplex were IFNγ, IL-1α, IL-6, IL-10, IL-12 p40, IL-12 p70, IL-15, IP-10, MCP-1, MIP-1β, and TNFα. Results are shown for cytokines that were observed to have variation between groups. pDNA was delivered on days 0, 4, and 7. Data are represented as the mean expression for each group. (**A**) levels of IL-1α; (**B**) levels of IL-6; (**C**) levels of IFNγ; (**D**) levels of MIP-1β. Error bars represent the standard deviation. *n* = 4 for each group.

**Figure 5 cancers-12-03072-f005:**
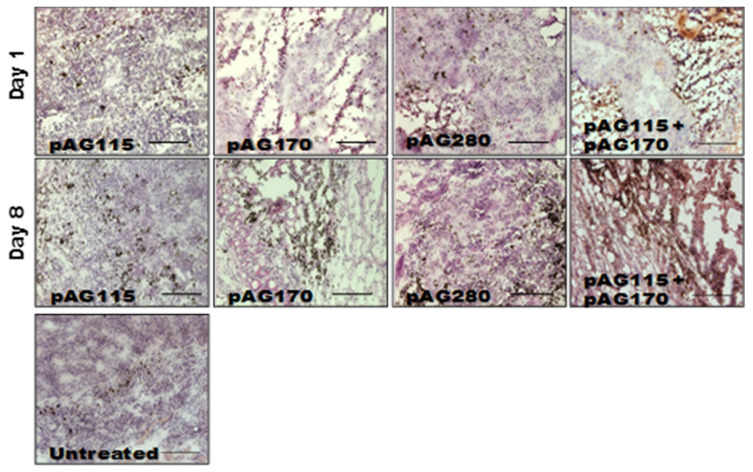
Histology of tumors after electrotransfer. Hematoxylin and eosin stained sections of tumor collected on day 1 (24 h after the first treatment) and on day 8 (24 h after the third treatment) after the electrotransfer of pAG115, pAG170, and pAC208 using EP2. Images are representative sections of *n* = 2 tumors. Scale bar = 500 µm.

**Figure 6 cancers-12-03072-f006:**
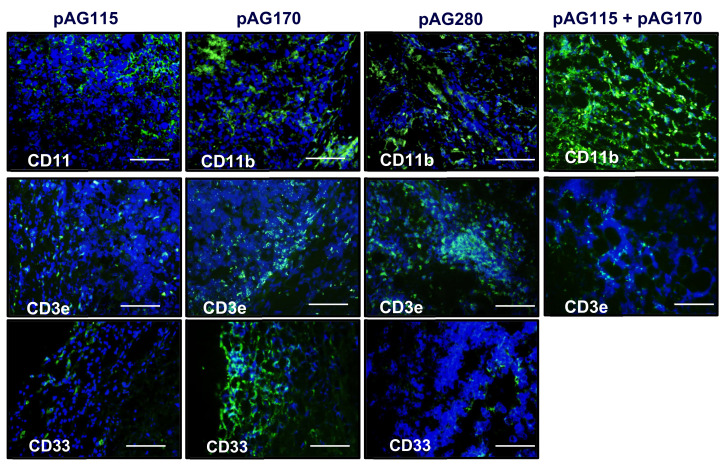
IL-15 GET induces immune cell infiltrate into the tumor microenvironment. Haematoxylin and eosin stain. Anti-CD3e (green-T cells), anti-CD11b (green-APCs), and anti-CD335 (green—natural killer cells) staining (green) reveal the presence of immune cells within the tumors of mice treated using EP2 to deliver pAG115, p,G170, and pAG208 on days 0, 4, and 7. Tissue was collected on day 8 (24 h after the third treatment). Images are the best representative sample of *n* = 2 for each group. Sections were counterstained with a nuclear dye 4′,6-diamidino-2-phenylindole (DAPI) (Blue). Mag = 20×. Scale bar = 500 µm.

**Figure 7 cancers-12-03072-f007:**
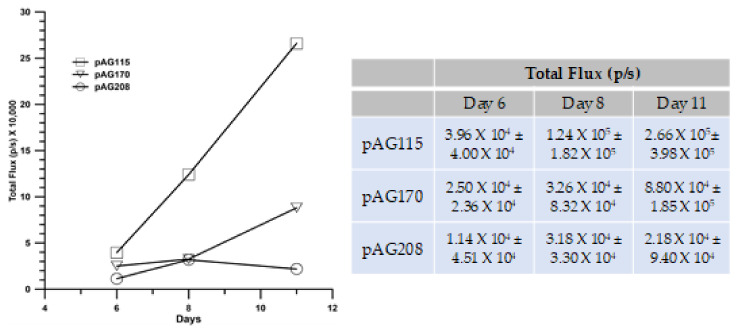
IL-15/IL-15Rα GET protects mice from secondary tumor formation in a lung colonization model. B16.F10 tumors were induced on the flanks of C57BL/6 Albino mice and treated with GET using the EP2 protocol on days 0, 4, and 7. On the day of first treatment, B16.F10-Luc G5 cells were injected intravenously. Tumor metastasis was monitored using an IVIS spectrum and reported as luminescence (photons/sec). *n* = 9 for each group.

**Table 1 cancers-12-03072-t001:** Long term survival of mice that received gene electrotransfer.

Plasmid	GET Protocol	Original	Challenged (Disease Free at 50 Days)	Resistant	% Resistant
		*n*	*n*	*n*	
pAG115 (IL-15Rα)	Injection Only	10	0	-	-
pAG208 (IL-15/IL-15Rα)	Injection Only	10	0	-	-
pAG170 (IL-15)	Injection Only	10	0	-	-
pAG170 + pAG115)	Injection Only	10	0	-	-
pAG115 (IL-15Rα)	EP1	10	0	-	-
pAG208 (IL-15/IL-15Rα)	EP1	10	6	4	40
pAG170 (IL-15)	EP1	10	9	3	30
pAG170 + pAG115	EP1	10	10	2	20
pAG115 (IL-15Rα)	EP2	10	2	0	0
pAG208 (IL-15/IL-15Rα)	EP2	10	9	7	70
pAG170 (IL-15)	EP2	10	7	0	0
pAG170 + pAG115	EP2	10	6	0	0

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
