# Peer review of "Electrotransfer of IL-15/IL-15Rα Complex for the Treatment of Established Melanoma"

_cancers, 2020, doi:10.3390/cancers12103072_

Round 1

Reviewer 1 Report

Clarity of the text needs improvement. Eg L 31 is barely understandable. L 50-51 makes no sense as written. L 32-33 correction - IL-15 is not a target of an anti-cancer agent. Too many names for the same thing: IL-15Ra = unique receptor alpha = the high affinity receptor; pAGXX = AGXX. L 109 was --> were. decide what tense to use then be consistent.

Need better description of IL-15 & IL-15Ra in intro, which should include reference to previous work and how this study extends those previous studies - eg Rowley 2009 is not referenced yet they put IL-15 & IL-15Ra on same plasmid and did tumor regression studies. L 65 authors reference their previous studies but no citation. 

Listing each confusing passage will be exhaustive. In summary, both the text and the figures are presented in an unclear way. One must read and re-read certain passages then examine the figures looking for clarification, which unfortunately were not helpful.

Reviewers typically look at figures first and then read the text for context and logical flow. Here, examination of figures and figure legends did not give a clear summary of results. The accompanying text was also not that useful. Eg L 124 says there is a critical correlation between electrotransfer and tumor regression but they don't say what that correlation is. Fig.1 X-axis: what are "N" and "C"? Fig 4 they measure which other cytokines are upregulated by the treatment, using a cytokine array, but neither the figure nor text says which cytokines were assayed. 

Conclusion? I'm left wondering: 1) why moderate levels of IL-15 worked better than high; 2) why some conditions did not affect tumor regression but did provide protection from challenge; and 3) on what levels do these results agree/disagree with those who did similar expts without the electrotransfer.

Author Response

Reply: We would like to thank the reviewers for the careful and thorough reading of our manuscript and for the thoughtful comments and constructive suggestions, which have helped to improve the quality of this manuscript. Our response is as follows.

Reviewer 1

Clarity of the text needs improvement. Eg L 31 is barely understandable. L 50-51 makes no sense as written. L 32-33 correction - IL-15 is not a target of an anti-cancer agent. Too many names for the same thing: IL-15Ra = unique receptor alpha = the high affinity receptor; pAGXX = AGXX. L 109 was --> were. decide what tense to use then be consistent.

Reply: Thank you for the suggestion. The text has been modified as per the suggestion

Need better description of IL-15 & IL-15Ra in intro, which should include reference to previous work and how this study extends those previous studies - eg Rowley 2009 is not referenced yet they put IL-15 & IL-15Ra on same plasmid and did tumor regression studies. L 65 authors reference their previous studies but no citation. 

Reply: Rowley, 2009 did not use plasmid DNA, they used RNA and did the delivery in vitro to inactivated T-cells. They had a study published in 2008, which used retrovirus to transduce tumor cells to express IL-15/IL-15Rα. These cells after injection into mice delayed tumor growth. IN contrast our study treated established tumors in vivo. While these papers did not delivery the genes or RNA in vivo, which one the focus of the current study, these references were added to the introduction.

Listing each confusing passage will be exhaustive. In summary, both the text and the figures are presented in an unclear way. One must read and re-read certain passages then examine the figures looking for clarification, which unfortunately were not helpful.

Reply: manuscript was carefully edited.

Reviewers typically look at figures first and then read the text for context and logical flow. Here, examination of figures and figure legends did not give a clear summary of results. The accompanying text was also not that useful. Eg L 124 says there is a critical correlation between electrotransfer and tumor regression but they don't say what that correlation is. Fig.1 X-axis: what are "N" and "C"? Fig 4 they measure which other cytokines are upregulated by the treatment, using a cytokine array, but neither the figure nor text says which cytokines were assayed. 

Reply: Thank you for the comment, we apologize for your confusion. This information has been clarified in the figures and text. It would be exhaustive to state or summarize all of the changes.

Conclusion? I'm left wondering: 1) why moderate levels of IL-15 worked better than high; 2) why some conditions did not affect tumor regression but did provide protection from challenge; and 3) on what levels do these results agree/disagree with those who did similar expts without the electrotransfer.

Reply: 1) immunotherapy achieves success by finding a dose that can induce the appropriate stimulation. When doses are too high it is possible to actually induce suppression or to exhaust immune cells or upregulate checkpoints. In addition, with high doses, it is possible that an innate response is stimulated and not an adaptive one. In this case you may see a local response but not stimulation of a memory response or one at a distant untreated site. 2) at no point in this manuscript did we suggest or have data that showed you can achieve protection from challenge if there was not a response in the treated tumor. Mice that did not achieve tumor regression did not survive long enough (50 days) to be challenged with reinjection of B16.F10 tumor cells. 3) our study does agree with other similar studies. Many studies do not treat established tumors, but try to block growth (i.e. studies you mentioned) or they report a slowing of tumor growth when treating established tumors. Also, not all studies perform a challenge with fresh injection of tumor cells after obtaining complete regression of primary treated tumor. In those cases, it is difficult to confirm immunostimulation or a memory response. 

Reviewer 2 Report

The present work describes how the intratumoral administration of IL-15, achieved by GET, can be used as immunotherapeutic approach in melanoma. The topic is very interesting since the considered disease is one of the best suited for immunotherapy and the analysed molecule has been widely investigated in cancer immunology field for its capabilities to boost immunity without the collateral effects observed for IL-2. So, in general, I would recommend the publication. However, there are some points that I think should be improved before the publication.

I listed here minor typos error that should be corrected. I would recommend the authors to carefully read again the manuscript, paying particular attention to singular/plural forms and related verbs:

Line 51: it should sound “aspects of this immunotherapy approach were exam

Lines 63-65: the reference should be included

Line 79: it should sound: “The two GET protocols utilized were EP1 and EP2

Line 86: before talking about the treatment, the authors should briefly mention the used system (mouse strain and sex and the tumor cells used for the challenge)

Lines 89-90: I would suggest the authors to reverse the order in which delivery and expression are mentioned, since usually you first release the plasmid and then assess the expression. In the present form, the overall meaning of the two sentences is quite confused

Line 100: it should sound “pAG115 are interesting as they support”

Line 108: it should sound “or when both pAG170 and pAG115 were delivered”

Line 110: it should sound “but they dropped”

Lines 105-106: it should sound “These data do not suggest a clear advantage for selecting to deliver IL-15 with IL-15Rα rather than IL-15 alone”

Lines 114-126: Figure 2 is not mentioned in the text

Line 130: it should sound “data are shown”

Line 137: it should sound “that were tumor-free”

Lines 152-153: it should sound “From these results it appears that different combinations of plasmids and electrotransfer parameters may induce”

Lines 154-156: It is also important to note that delivery of pAG208 with EP2 did not result in high levels of IL-15 or IL-15/IL-15Rα complex. However, it still did result in the highest level of protection from challenge”

Line 256: it should sound “was shown to be”

Line 262: it should sound “nor induces activation-induced cell death”

Line 289: it should sound “Though there was not an apparent advantage in using”

Line 318: I think that, after “saline”, there is a missing word (buffer? Solution?)

Lines 321-322: it should sound “Plasmids pAG170 and pAG115 express IL-15 with a CM-CSF secretory signal sequence and IL-15Rα from a human CMV promoter, respectively.”

Lines 336-338: it should sound “Mice whose tumor volume exceeded 1000 mm3 were removed from the study and humanely euthanized.”

Line 380: it should sound “data were”

The discussion is overall quite exhaustive in describing the clinical usage of IL-15 and the limited effectiveness of the therapy due to limited half life and bioavailability of the molecule. However, the authors does not mention the negative effects induced by high doses of IL-15, which are in line with their results, nor discuss why and how high levels of IL-15 can impact immune response. I report here two useful references the authors could use: a review mentioning the double-sword effect of IL-15 therapy, particularly on NK cells (Cristiani et al. Scand J Immunol. 2020 Apr;91(4):e12861. doi: 10.1111/sji.12861.) and a paper describing the negative role of IL-15 in predicting the response to ipilimumab and its effect on NK and T cell phenotype (Tallerico et al. Oncoimmunology. 2016 Dec 7;6(2):e1261242. doi: 10.1080/2162402X.2016.1261242)

I have a concern about the statistical approach used to analyse data. Mann-Whitney test is generally used for unrelated groups of samples (i.e. healthy donors vs. patients). For pair-wise comparisons, Wilcoxon signed-rank test or paired t-test, based on the samples distribution, are recommended

In my opinion, figures require some revision. In general, I would recommend:

- Statistic tests used should be mentioned in all the captions

- The difference between EP1 and EP2 (needle vs. caliper) is clarified in the text, so the authors should avoid to specify it into figure legends because it make graphs more confused. If they want, they can specify this difference in the captions

About specific figures:

- Figure 1: the graphs are not very clear. I do not understand why plasmid administration without GET is reported only for AG208 and not also for the other plasmids/combinations: the authors should clarify this point. About the figure, the font is difficult to read, I would suggest to improve the size. The columns are also very small. In order to improve the figure, the authors could put the graphs in column instead of in line: in this way they could enlarge the columns and use a single X-axis and a single legend applied to both the graphs

- Figures 2 and 3 are both very difficult to read. I would suggest to increase the size of fonts, lines and symbols and to use a huger set of symbols (full/empty circles, squares, triangles, diamonds instead to differently oriented triangles)

- Figures 5 and 6: please mention the mouse strain

Author Response

Reply: We would like to thank the reviewers for the careful and thorough reading of our manuscript and for the thoughtful comments and constructive suggestions, which have helped to improve the quality of this manuscript. Our response is as follows.

Reviewer 2

The present work describes how the intratumoral administration of IL-15, achieved by GET, can be used as immunotherapeutic approach in melanoma. The topic is very interesting since the considered disease is one of the best suited for immunotherapy and the analysed molecule has been widely investigated in cancer immunology field for its capabilities to boost immunity without the collateral effects observed for IL-2. So, in general, I would recommend the publication. However, there are some points that I think should be improved before the publication.

Reply: Thank you

 I listed here minor typos error that should be corrected. I would recommend the authors to carefully read again the manuscript, paying particular attention to singular/plural forms and related verbs:

Reply: all suggested changes have been made except for Lines 63-65: the reference should be included”. In this case we were referring to this current study. This was modified to remove the results so as not to be confusing.

The discussion is overall quite exhaustive in describing the clinical usage of IL-15 and the limited effectiveness of the therapy due to limited half life and bioavailability of the molecule. However, the authors does not mention the negative effects induced by high doses of IL-15, which are in line with their results, nor discuss why and how high levels of IL-15 can impact immune response. I report here two useful references the authors could use: a review mentioning the double-sword effect of IL-15 therapy, particularly on NK cells (Cristiani et al. Scand J Immunol. 2020 Apr;91(4):e12861. doi: 10.1111/sji.12861.) and a paper describing the negative role of IL-15 in predicting the response to ipilimumab and its effect on NK and T cell phenotype (Tallerico et al. Oncoimmunology. 2016 Dec 7;6(2):e1261242. doi: 10.1080/2162402X.2016.1261242)

Reply: thank you for the suggestions. The discussion was modified to include these references.

I have a concern about the statistical approach used to analyse data. Mann-Whitney test is generally used for unrelated groups of samples (i.e. healthy donors vs. patients). For pair-wise comparisons, Wilcoxon signed-rank test or paired t-test, based on the samples distribution, are recommended

Reply: thank you for the comment. We agree with you and have modified the manuscript accordingly.

 In my opinion, figures require some revision. In general, I would recommend:

- The difference between EP1 and EP2 (needle vs. caliper) is clarified in the text, so the authors should avoid to specify it into figure legends because it make graphs more confused. If they want, they can specify this difference in the captions

Reply: thank you for the suggestion, the changes were made.

About specific figures:

- Figure 1: the graphs are not very clear. I do not understand why plasmid administration without GET is reported only for AG208 and not also for the other plasmids/combinations: the authors should clarify this point. About the figure, the font is difficult to read, I would suggest to improve the size. The columns are also very small. In order to improve the figure, the authors could put the graphs in column instead of in line: in this way they could enlarge the columns and use a single X-axis and a single legend applied to both the graphs

- Figures 2 and 3 are both very difficult to read. I would suggest to increase the size of fonts, lines and symbols and to use a huger set of symbols (full/empty circles, squares, triangles, diamonds instead to differently oriented triangles)

- Figures 5 and 6: please mention the mouse strain

Reply: thank you for the comments and suggestions, these changes were made.

Round 2

Reviewer 1 Report

Thank you for revising the manuscript and responding to comments. It was a pleasure to read the revised manuscript. The importance of several of the findings is now very clear.